A novel model based on clinical and computed tomography (CT) indices to predict the risk factors of postoperative major complications in patients undergoing pancreaticoduodenectomy

Wang Jiaqi 1
Xu Kangjing 1
http://orcid.org/0009-0009-8227-1798 Zhou Changsheng 2
Wang Xinbo 1
Zuo Junbo 3
Zeng Chenghao 1
Zhou Pinwen 1
Gao Xuejin 1
Zhang Li 1
Wang Xinying 1 wangxinying@nju.edu.cn
1 Department of General Surgery, Nanjing Jinling Hospital, Affiliated Hospital of Medical School, Nanjing University , Nanjing , China
2 Department of Radiology, Nanjing Jinling Hospital, Affiliated Hospital of Medical School, Nanjing University , Nanjing , China
3 Department of General Surgery, The Affiliated People’s Hospital of Jiangsu University , Zhenjiang , China
Zhang Xin
Electronic publication date: 2024 Dec 19
Publication date: 2024
Volume: 12
Electronic Location ID: e18753
Received 2024 Jun 18; Accepted 2024 Dec 3
Copyright: © 2024 Wang et al.
Copyright year: 2024
Copyright holder: Wang et al.
License: This is an open access article distributed under the terms of the Creative Commons Attribution License, which permits unrestricted use, distribution, reproduction and adaptation in any medium and for any purpose provided that it is properly attributed. For attribution, the original author(s), title, publication source (PeerJ) and either DOI or URL of the article must be cited.
License URL: https://creativecommons.org/licenses/by/4.0/

Keywords: Pancreaticoduodenectomy, Postoperative complications, Three-dimensional CT reconstruction, Predictive model

Funding: National Natural Science Foundation of China 82170575, 82370900 This work was supported by the National Natural Science Foundation of China (82170575, 82370900). The funders had no role in study design, data collection and analysis, decision to publish, or preparation of the manuscript.

==============================
Background

Postoperative complications are prone to occur in patients after radical pancreaticoduodenectomy (PD). This study aimed to construct and validate a model for predicting postoperative major complications in patients after PD.

Methods

The clinical data of 360 patients who underwent PD were retrospectively collected from two centers between January 2019 and December 2023. Visceral adipose volume (VAV) and subcutaneous adipose volume (SAV) were measured using three-dimensional (3D) computed tomography (CT) reconstruction. According to the Clavien-Dindo classification system, the postoperative complications were graded. Subsequently, a predictive model was constructed based on the results of least absolute shrinkage and selection operator (LASSO) multivariate logistic regression analysis and stepwise (stepAIC) selection. The nomogram was internally validated by the training and test cohort. The discriminatory ability and clinical utility of the nomogram were evaluated by area under the receiver operating characteristic (ROC) curve (AUC), calibration curve, and decision curve analysis (DCA).

Results

The major complications occurred in 13.3% (n = 48) of patients after PD. The nomogram revealed that high VAV/SAV, high system inflammation response index (SIRI), high triglyceride glucose-body mass index (TyG-BMI), low prognostic nutritional index (PNI) and CA199 ≥ 37 were independent risk factors for major complications. The C-index of this model was 0.854 (95%CI [0.800–0.907]), showing excellent discrimination. The calibration curve demonstrated satisfactory concordance between nomogram predictions and actual observations. The DCA curve indicated the substantial clinical utility of the nomogram.

Conclusion

The model based on clinical and CT indices demonstrates good predictive performance and clinical benefit for major complications in patients undergoing PD.

Introduction

Pancreatoduodenectomy (PD), acknowledged as a classic surgical method, is commonly performed for malignant pancreatic tumors, adenocarcinoma of ampulla, duodenal tumor, cholangiocarcinoma and some benign tumors. Although with the improvement of the PD and the standardization of perioperative management, operative mortality, and morbidity rates have decreased dramatically, postoperative complication rates still vary from 20% to 60% (Menahem et al., 2015). Common complications after PD comprise pancreatic fistula, bile leakage, intestinal fistula, abdominal infection, pulmonary infection, postpancreatectomy hemorrhage (PPH), delayed gastric emptying (DGE), and organ dysfunction (Karim et al., 2018). Such complications would ultimately prolong hospital stays, increase hospitalization costs, and even lead to readmission in the post-discharge period. Major complications, typically classified as Clavien-Dindo grade ≥3, significantly impact the short-term outcomes of PD and may also negatively influence long-term survival according to the expert consensus (Study Group of Pancreatic Surgery in China Society of Surgery of Chinese Medical Association, Pancreatic Disease Committee of China Research Hospital Association, and Editorial Board of Chinese Journal of Surgery, 2023). Therefore, correctly evaluating the occurrence of major complications after PD is of great significance for guiding clinical practice. To further decrease the incidence of postoperative complications after PD, there is a clinical need to clarify potential risk factors in the perioperative period. According to results, clinical doctors can intervene early and ultimately improve postoperative clinical outcomes.

Currently, researches showed that advanced age (Faraj et al., 2013), a high subcutaneous adipose (SAT) area in the paralumbar section of the third lumbar spine (L3) (van Dijk et al., 2017), and high body mass index (BMI) (Shen et al., 2021a) are considered as risk factors for postoperative morbidity and mortality following PD (Matsui et al., 2022; Watanabe et al., 2012). Moreover, other clinical indicators including preoperative nutritional condition, hyperglycemia, hyperlipidemia, and inflammatory status can also increase the risk (Chen et al., 2022; Gouillat & Gigot, 2001; Shi et al., 2023). However, BMI and L3 VAT cannot adequately reflect the fat distribution of the body. Visceral adipose volume (VAV) and subcutaneous adipose volume (SAV) at the L3 vertebral level were measured using three-dimensional (3D) computed tomography (CT) reconstruction technology to calculate the VAV/SAV ratio in patients. This approach was applied to provide a more accurate representation of body composition in the patient cohort. The triglyceride glucose-body mass index (TyG-BMI), derived from the combination of the TyG index with obesity indices, reflects the blood glucose, lipid levels, and insulin resistance, which is closely associated with various postoperative complications (Yang et al., 2023). Recent studies showed that the occurrence and development of pancreatic cancer are directly related to high insulin levels (Zhang et al., 2023). Consequently, we explored the value of the TyG-BMI index in predicting the risk of major complications in patients undergoing PD.

Additionally, recent studies have highlighted the significant role of inflammation and nutritional status in influencing the outcomes of PD (Gilliland et al., 2017; Ma et al., 2023). Preoperative inflammation, often driven by systemic immune responses to malignancy, and poor nutritional status have both been linked to increased rates of postoperative complications. The system inflammation response index (SIRI) is a novel, non-invasive inflammatory biomarker that comprehensively reflects the balance between the host’s immune and inflammatory states (Geng et al., 2018). The prognostic nutritional index (PNI), a simple and effective indicator reflecting the nutritional status of the body, has been demonstrated in multiple studies to have predictive value for the prognosis of gastrointestinal malignancies (Ding et al., 2022). However, there have been relatively few studies exploring their impact on the occurrence of major complications after PD. Therefore, we have also included them in our analysis.

The nomogram is a statistical tool for predicting individual-specific outcomes by transforming complex regression equations into visual graphs. Numerous studies have demonstrated the clinical utility of the nomogram in predicting complications and prognosis after PD (Zhang et al., 2022b; Zhu et al., 2022). Therefore, this study aimed to develop a novel nomogram based on clinical and CT indices to predict the risk factors of patients after PD on the basis of preoperative clinical indicators that can be easily obtained in routine clinical practice.

Methods

Study population

The study population were retrospectively enrolled between January 2019 and December 2023 from two centers, Jinling Hospital and The Affiliated People’s Hospital of Jiangsu University. The flowchart of this study is shown in Fig. S1.

All patients admitted would first receive an assessment of surgical resectability by the multidisciplinary team (MDT) through imaging to determine curative resectability of tumors. The inclusion criteria are all patients that underwent upfront PD during the study period at the designated institutions by expert pancreatic surgeons. Finally, a total of 360 patients were included in the study. Subsequently, their medical records, laboratory results, and imaging data of these patients were collected and systematically organized for further analysis. In terms of the Clavien-Dindo system, the cohort was divided into those with or those without major complications. Written informed consent was obtained from patients prior to the research. This study conformed to the ethical guidelines of the 1975 Declaration of Helsinki and was approved by the Ethics Committee of Jinling Hospital (2023DZKY-049-03) and registered at ClinicalTrials.gov (NCT06378853).

Surgical procedures and perioperative management

All patients underwent an MDT discussion before treatment. The open Kausch-Whipple procedure is performed according to the NCCN guidelines (Tempero et al., 2021). The surgical procedure generally includes: opening the Kocher incision, organ resection, anastomotic reconstruction in hand-sewn by continuous suturing techniques, and irrigating the peritoneal cavity.

Drugs inhibiting secretion of digestive fluid such as proton-pump inhibitors and somatostain were administered routinely. Laboratory results, including routine blood and biochemical examinations, were collected on postoperative day (POD) 1. All patients in the study underwent abdominal CT or trans-abdominal ultrasound within the first week after surgery. Additional abdominal imaging was conducted when there was a sign of possible intra-abdominal complications. Oral diet, walking, and early withdrawal of drainage tubes were advocated early postoperatively. Liquid diet was gradually resumed around POD 2–5 and soft diet after defecation. Drainage tubes were removed appropriately on day 3 and day 5 after surgery if there were no signs of leakage and drainage fluid amylase levels were within normal range.

Data collection

The clinical information of all patients included age, sex, and BMI, diabetes, hypertension and ASA. The preoperative data closest to the time of surgery were collected for analysis, including white blood cell (WBC) count, red blood cell (RBC) count, platelet (PLT) count, hemoglobin (Hb) level, albumin (ALB) level, lymphocyte count, monocyte count, neutrophil count, triglyceride (TG) level, fasting plasma glucose (FPG) level, and CA199 level. The prognosis nutrition index (PNI) was calculated using the formula: PNI = serum ALB level (g/L) + 5 × total number of peripheral blood lymphocytes (×109/L) (Onodera, Goseki & Kosaki, 1984), and the systemic inflammation response index (SIRI) was calculated using the formula: SIRI = total number of peripheral blood neutrophils (×109/L) × total number of peripheral blood monocytes (×109/L)/total number of peripheral blood lymphocytes (×109/L) (Qi et al., 2016). The TyG index was calculated using the formula: TyG = Ln [TG (mg/dl) × FPG (mg/dl)/2] (Simental-Mendia, Rodriguez-Moran & Guerrero-Romero, 2008). The TyG-BMI index was defined as the TyG index multiplied by BMI (Kg/m2). The postoperative data such as maximum tumor size, length of hospital stay, and postoperative complications were analyzed. Postoperative complications were mainly evaluated based on the diagnostic criteria established by the International Study Group of Pancreatic Surgery (ISGPS) (Bassi et al., 2017; Koch et al., 2011; Wente et al., 2007a, 2007b).

Collection of CT data

A single trained radiologist analyzed CT images preoperatively using the HY-QCT medical software version V2.5.0. The software is based on high-quality CT imaging data and simulates the human visual nervous system’s method of detecting target objects in images through convolutional neural networks. It trains the computer to become a perceptual system composed of neurons that are extremely sensitive to the features of the spinal region in CT images. By processing, analyzing, and calculating the spinal region in CT images, it achieves segmentation detection of spinal vertebrae, morphological analysis of vertebrae, and tissue analysis. Then, the tissue composition, such as L3 SAV, VAV, abdominal wall muscular spatium fat, paravertebral muscular spatium fat, abdominal wall muscle, and paraspinal muscle were segmented automatically and their volume were calculated (Fig. 1).

Figure 1 3D CT reconstruction to analyze the L3 adipose and muscle tissue.

(A) Analysis results of patient’s adipose and muscle tissue. (B) Axial slice reformation of a preoperative CT scan at the L3 level. (C) Sagittal reformation at the L3 level of the same patient. (D) Coronal plane reformation at the L3 level of the same patient.

Definition of cut-off value

The BMI ≤ 18.5 kg/m2 indicates that weight is below the healthy range, suggesting the possibility of malnutrition, chronic illness, or other health issues according to clinical management and treatment of obesity in China (Zeng et al., 2021). CA199 ≥ 37 U/ml indicates that it exceeds the normal healthy range, suggesting the possibility of pancreatic cancer or other gastrointestinal tumors based on comprehensive guidelines for the diagnosis and treatment of pancreatic cancer (Pancreatic Cancer Committee of Chinese Anticancer A, 2021). Due to the lack of specific cut-off values in previous studies for VAV/SAV, SIRI, PNI, and TyG-BMI, we determined the optimal cut-off value for the classification performance of these indicators using the area under the receiver operating characteristic (ROC) curves. The optimal cut-off value was determined by maximizing specificity and sensitivity using Youden’s Index. Therefore, SIRI ≥ 0.86 was defined as high SIRI, PNI ≤ 47.65 as low PNI, TyG-BMI ≥ 200.11 as high TyG-BMI, and VAV/SAV ≥ 1.59 as high VAV/SAV.

Feature selection and nomogram development

From all demographic features and preoperative clinical data, to prevent overfitting and handle multicollinearity, these variables which showed differences between the two groups (P < 0.05) and other potential risk factors were evaluated by the LASSO regression. Then it produced 11 significant parameters (age ≥ 65, diabetes, hypertension, CA199 ≥ 37, NWR, SIRI, PNI, TyG-BMI, VAV/SAV, jaundice and Hb that had nonzero coefficients. And we obtained the lambda min value when the MSE is minimized, which indicates that the prediction error of the model is at its lowest.).

Then those significant parameters were inputted into the final multivariate regression analysis. The multivariate regression analysis conducted as described demonstrated that CA199 ≥ 37, high SIRI, low PNI, high TyG-BMI, and high VAV/SAV were independent risk factors for major complications after PD. Next, the stepAIC selection was performed to obtain the best prediction model. Based on the results, a nomogram model was established to predict major complications after PD.

Statistical analysis

Continuous variables with a normal distribution are presented as the mean and standard deviation (SD) and were analyzed using the Student’s t-test. Continuous variables with a non-normal distribution are presented as the median with interquartile range (IQR), and the Mann-Whitney U test was used for analysis between the two groups. Chi-squared or Fisher test was used to compare the distribution of categorical variables between groups. Two-sided p-values less than 0.05 were considered significant. All statistical analyses were conducted using IBM SPSS Statistics for Windows, version 25.0 (IBM Corp., Armonk, NY, USA) and R version 4.0.5 (R Core Team, 2021).

The model building followed a predefined plan. Firstly, data cleaning and imputation were conducted for datasets. Variables with missing percentages of more than 10% were deleted, and other variables were filled with multiple imputation methods (Mice package in R). Secondly, least absolute shrinkage and selection operator analysis was used to filter valuable diagnostic variables. Next, multivariate logistic regression analyses were used to determine independent risk factors for major complications in patients undergoing PD. StepAIC selection was then performed to obtain the best prediction model. Nomograms were constructed based on the proportional conversion of each regression coefficient from multivariate logistic regression onto a 0- to 100-point scale. Then, the nomogram was internally validated by the training and test database.

The optimal cut-off value of the nomogram score was calculated using the maximum Youden index method. The area under the curve (AUC) was subsequently corrected by bootstrapping validation (1,000 bootstrap resamples) (Harrell, Lee & Mark, 1996; Steyerberg et al., 2001). The Cox-Snell R2, Nagelkerke R2 and Hosmer-Lemeshow R2 (HL R2) were used to fit of the model. The calibration curve, and decision curve analysis (DCA) were used to evaluate this model.

Results

Clinicopathological characteristics of patients and incidence of postoperative complications

Of 415 PDs performed at the participating institutions during the study period, 360 (86.7% in relation to 415) met the criteria for analysis. The clinical pathological characteristics of these patients were shown in Table 1. There were 173 males (48.1%) and 187 females (51.9%). The median age of the patients was 61.0 ± 10.6 years. Among baseline characteristics, significant differences were noted for age, diabetes BMI, lymphocyte, neutrophil, FPG, TG, and CA199 between the two groups. Tumor size and duration of hospital stay differed significantly between the two groups.

Table 1 Patient demographics and clinical characteristics are divided by with or without major complications.

Variables	Total	Without major
complications	With major
complications	p value	
Total number	360	312	48		
Age, year	61.0 ± 10.6	60.4 ± 10.9	64.5 ± 7.6	0.013*	
Sex, female/male				0.066	
Female	187	168	19		
Male	173	144	29		
BMI ≤ 18.5 kg/m2, n	45	34	11	0.019*	
Diabetes				0.041	
No	255	227	28		
Yes	105	85	20		
Hypertension				0.145	
No	250	221	29		
Yes	110	91	19		
Jaundice				0.664	
No	235	205	30		
Yes	125	107	18		
ASA				0.130	
1	143	128	15		
2	183	158	25		
≥ 3	34	26	8		
WBC, 109/L	6.54 (5.20–7.30)	6.52 (5.04–7.27)	6.90 (5.92–7.35)	0.059	
RBC, 1012/L	3.96 (3.79–4.15)	3.96 (3.79–4.15)	3.95 (3.78–4.14)	0.639	
PLT, 109/L	202 (186–212)	202 (186–215)	200 (187–207)	0.151	
Hb, g/L	123 (113–135)	123 (113–137)	121 (110–131)	0.156	
Lymphocyte, 109/L	1.59 (1.22–1.95)	1.63 (1.24–1.98)	1.37 (1.08–1.68)	0.002**	
Monocyte, 109/L	0.47 (0.38–0.60)	0.47 (0.37–0.60)	0.50 (0.41–0.61)	0.142	
Neutrophil, 109/L	3.95 (2.93–5.20)	3.90 (2.78–5.29)	4.12 (3.63–5.10)	0.037*	
NLR	2.59 (1.84–3.32)	2.51 (1.73–3.21)	3.07 (2.43–4.42)	<0.001***	
NWR	0.64 (0.56–0.72)	0.63 (0.55–0.72)	0.65 (0.57–0.78)	0.200	
MWR	0.08 (0.06–0.10)	0.08 (0.06–0.10)	0.08 (0.06–0.09)	0.556	
High SIRI, n	252	206	46	<0.001***	
ALB ≤ 35 g/L, n	65	51	11	0.262	
FPG, mmol/L	4.50 (3.90–5.40)	4.42 (3.90–5.30)	5.10 (4.46–5.62)	0.011*	
TG, mmol/L	1.72 (1.36–2.08)	1.63 (1.30–1.91)	2.09 (1.95–2.27)	<0.001***	
Low PNI, n	172	131	41	<0.001***	
High TyG-BMI, n	53	37	16	<0.001***	
CA199 ≥ 37 U/ml, n	153	126	27	0.038*	
VAV, cm3	3,587.90 (2,413.52–4,739.48)	3,415.08 (2,310.62–4,593.01)	4,238.64 (3,100.14–5,360.96)	0.004**	
SAV, cm3	3,470.36 (2,400.66–4,512.26)	3,620.71 (2,400.66–4,818.90)	2,941.49 (2,379.68–3,885.39)	0.032*	
Abdominal wall muscle volume, cm3	1,608.5 (1,310.1–1,955.9)	1,610.5 (1,291.8–1,943.5)	1,607.0 (1,364.1–1,999.7)	0.631	
Paraspinal muscle volume, cm3	1,955.8 (1,618.1–2,456.4)	1,958.5 (1,613.2–2,471.8)	1,929.9 (1,647.8–2,386.3)	0.918	
Total muscle volume, cm3	3,567.0 (2,860.6–4,360.4)	3,571.0 (2,844.5–4,356.8)	3,527.0 (2,946.3–4,406.3)	0.791	
High VAV/SAV, n	74	48	26	<0.001***	
Tumor size, (cm, n)				<0.001***	
<3	178	165	13		
≥3	182	147	35		
Duration of hospital stay, day	21.3 ± 9.3	19.8 ± 8.2	31.2 ± 10.5	<0.001***	
Pancreatic texture, n					
soft	256	224	32	0.466	
hard	104	88	16		
Pancreas duct size, (mm, n)				0.563	
≤3	179	157	22		
>3	181	155	26		
Operative time, min	300.0 (280.0–320.0)	300.0 (280.0–320.0)	305.0 (282.5–327.5)	0.099	
Blood loss, ml	300.0 (250.0–350.0)	300.0 (250.0–350.0)	320.0 (262.5–390.0)	0.058	
Pathology				0.412	
PDAC	204	182	22		
Adenocarcinoma of ampulla	53	43	10		
Duodenal tumor	37	32	5		
Cholangiocarcinoma	32	25	7		
IPMN	22	20	2		
Others	12	10	2		
Notes:

* P < 0.05.

** P < 0.01.

*** P < 0.001.

BMI, body mass index; ASA, American Society of Anesthesiologists; WBC, white blood cell; RBC, red blood cell; PLT, platelet; Hb: hemoglobin; ALB, albumin; FPG, fasting plasma glucose; TG, triglyceride; NLR, neutrophil to lymphocyte ratio; NWR, neutrophil to white blood cell ratio; MWR, monocyte to white blood cell ratio; SIRI, systemic inflammation response index; PNI, prognosis nutrition index; TyG-BMI, triglyceride glucose-body mass index; VAV, visceral adipose volume; SAV, subcutaneous adipose volume; PDAC, pancreatic ductal adenocarcinoma; IPMN, intraductal papillary mucinous tumor.

All postoperative complications were present in Tables S1 and S2. There were 48 cases (13.3%) of major complications in the total cohort. A total of 53 (14.7%) patients experienced pancreatic fistula, 16 (4.4%) patients had bile leakage and 6 (1.7%) patients had intestinal fistula. The rates of abdominal and pulmonary infection were 14.2% and 6.9% respectively. The PPH occurred in 28 (7.8%) patients. The DGE and organ dysfunction rates were 9.2% and 2.5%, respectively.

Risk-associated factors of major complications after PD

The results of LASSO are shown in Figs. S2 and S3. Then those significant parameters were inputted into the multivariate regression analysis. The results of multivariate regression analysis for risk factor selection are as follows (Table 2). To obtain the best model, we used the stepAIC method to select variables: one variable is added at a time, but at each step, the variables are re-evaluated. Variables that do not contribute to the model will be removed. The AIC value is calculated to assess the quality of each candidate model, and this process continues until the optimal model is obtained. Cox-Snell R2, Nagelkerke R2 and HL R2 are commonly used metrics to assess the goodness of fit of the selected model. As demonstrated in Table 3, the goodness-of-fit statistics for this model outperform those of the alternative models, suggesting that it exhibits superior explanatory power, accuracy, and a better fit to the data. The nomogram model to predict major complications after PD was established according to these results shown in Fig. 2. The total score was calculated using VAV/SAV, PNI, SIRI, TyG-BMI, and CA199. Each of these variables was assigned a score based on a scaled axis. The individual scores were then summed to obtain the total score. By mapping this total score to the corresponding point scale, we were able to estimate the risk of major postoperative complications in patients after PD. Contribution of each risk factor to the prediction model were shown in Table 3. The AUC and C-index of the nomogram was 0.854 (95% CI [0.800–0.907]), the positive and negative predictive value were 0.700 and 0.883 respectively, and accuracy was 0.878, indicating that the model exhibited good discrimination and substantial accuracy in predicting major complications after PD (Fig. 3).

Table 2 Multivariate analyses of risk factors for major complications after PD.

Factor	Multivariate analysis	
β coefficient	SE	Wald	OR	95% CI	p value	
High SIRI	1.952	0.777	6.310	7.045	[1.536–32.319]	0.012*	
NWR	−0.394	0.839	0.220	0.675	[0.130–3.495]	0.639	
Low PNI	1.483	0.461	10.349	4.406	[1.785–10.876]	0.001**	
High TyG-BMI	1.349	0.452	8.907	3.852	[1.589–9.338]	0.003**	
High VAV/SAV	1.611	0.396	16.591	5.009	[2.307–10.877]	<0.001***	
Age ≥ 65	0.317	0.381	0.692	1.373	[0.650–2.901]	0.406	
CA199 ≥ 37	0.839	0.388	4.667	2.314	[1.081–4.954]	0.031*	
Jaundice	−0.376	0.399	0.888	0.687	[0.314–1.501]	0.346	
Hb	−0.011	0.012	0.724	0.989	[0.966–1.014]	0.395	
Diabetes	0.190	0.567	0.112	1.209	[0.398–3.677]	0.738	
Hypertension	0.138	0.563	0.060	1.148	[0.381–3.464]	0.806	
Notes:

* P < 0.05.

** P < 0.01.

*** P < 0.001.

SE, standard error; OR, odds ratio; CI, confidence interval; SIRI, systemic inflammation response index; NWR, neutrophil to white blood cell ratio; PNI, prognosis nutrition index; TyG-BMI, triglyceride glucose-body mass index; VAV, visceral adipose volume; SAV, subcutaneous adipose volume; Hb, hemoglobin.

Table 3 Contribution of variables to the prediction of major complications in patients.

Variables	AIC	Cox-Snell R2	Nagelkerke R2	HL R2	AUC	
High VAV/SAV	−813.39	0.084	0.155	0.112	0.694 (0.620–0.768)	
+Low PNI	−834.28	0.144	0.264	0.198	0.793 (0.724–0.862)	
+High TyG-BMI	−844.08	0.163	0.300	0.227	0.809 (0.746–0.872)	
+CA199 ≥ 37	−848.53	0.179	0.330	0.252	0.834 (0.777–0.891)	
+High SIRI	−851.54	0.198	0.365	0.281	0.854 (0.800–0.907)	
Note:

AIC, akaike information criterion; HL, Hosmer-Lemeshow test; AUC, area under the curve; VAV, visceral adipose volume; SAV, subcutaneous adipose volume; PNI, prognosis nutrition index; SIRI, systemic inflammation response index; TyG-BMI, triglyceride glucose-body mass index.

Figure 2 Nomogram to predict the probability of major complications after PD.

Figure 3 The ROC curve of the predictive model for major complications after PD.

Calibration and validation of the nomogram

The calibration curve via 1,000 bootstrap resampling was used to assess the predictive accuracy of the nomogram. The calibration curve for the probability showed good concordance between the predicted and actual observations and the absolute error between the simulated curve and the actual curve is 0.016 (Fig. 4), which demonstrated the nomogram had great accurate predictive ability. The DCA converts complex mathematical models into simple and easy-to-understand graphics to intuitively judge the practicability and net benefits of different models. Therefore, in order to further investigate the clinical utility of the prediction model, the DCA (Fig. 5) was drawn with the threshold probability as the abscissa and the net benefit rate of patients as the ordinate. It can be clearly shown that patients whose postoperative major complications were predicted were predicted based on the nomogram have a higher net benefit than any patient predicted by a single indicator, and the nomogram had a better potential for clinical application when the threshold probability was 0.10 to 0.80, indicating that the prediction model has great clinical value.

Figure 4 The calibration curve of the nomogram for predicting the probability of major complications (bootstrap = 1,000 repetitions).

Figure 5 DCA curves of the prediction model to predict the probability of major complications.

To test the applicability of our prediction model, patients were divided into the training database (n = 251) and test database (n = 109) randomly with a ratio of 7:3. Internal validation of this model showed that the AUC was 0.864 and data from the test database was used for cross validation of the model, showing the AUC of 0.824 (Fig. S4). The calibration curves of the training and test database also demonstrated that the prediction model was in great agreement with actual observations (Figs. S5, S6). The DCA curves also revealed that the model provided great clinical benefit in the training and test database (Figs. S7, S8).

Discussion

Due to all cause-morbidity rates after PD are reported to vary from 38% to 77%, whilst procedure specific complications vary between 13% to 50% (Bassi et al., 2017), we should emphasize the importance of recognizing and detecting postoperative complications early. Several studies (El Nakeeb et al., 2013; Mintziras et al., 2020; Nakata et al., 2013; Palumbo et al., 2021) have established some prediction models to predict the occurrence of postoperative complications after PD, but the major complications are one of the most concerning issues to surgeons. Because they influence short-term outcomes and may also impact long-term survival severely.

In this study, clinical data from patients who underwent PD at two centers were retrospectively analyzed to identify risk factors associated with major complications. Additionally, a nomogram predictive model was developed based on 3D CT reconstruction, offering clinicians an intuitive and simple tool to identify risk factors and implement active medical interventions for prevention. The multivariable regression analysis identified the preoperative CA199 ≥ 37, VAV/SAV, PNI, SIRI, and TyG-BMI as independent risk factors for major complications. Consequently, these parameters were incorporated into the nomogram predictive model. Further internal validation, calibration curve, and DCA curve showed this model had good predicting ability and great value in clinical application. Additionally, this represents the first prediction model based on 3D CT reconstruction which combines multiple comprehensive indicators simultaneously.

Previous studies have explored the impact of body composition on postoperative complications in patients undergoing abdominal surgery. The high VAT/SAT ratio was significantly associated with postoperative complications and survival (Matsui, Inaki & Tsuji, 2021; Pacquelet et al., 2022). Visceral obesity has already proven to be associated with complications such as anastomotic leakage and postoperative pneumonia (Ballian et al., 2012). However, most current investigators analyzed CT images through SliceOmatic software to calculate parameters such as L3 skeletal muscle tissue (cm2), VAT (cm2), and SAT area (cm2), which may not accurately reflect the true composition of patients. At the same time, traditional technology uses serial processing methods in CT image feature extraction. It is prone to loss of image data, which results in problems such as ring distortion of the reconstructed image and long reconstruction time. Therefore, we segmented each structure of interest based on convolutional neural networks to calculate parameters including abdominal wall muscle (cm3), paraspinal muscle (cm3), VAV (cm3), and SAV (cm3). This study found that high VAV/SAV ratio was significantly associated with postoperative complications after PD, which is consistent with the literature before. It is considered that excessive fat accumulation may lead to the generation of adipokines and the infiltration of pro-inflammatory macrophages, thereby increasing the risk of postoperative complications (Deng et al., 2016).

Similar to high VAV/SAV, studies have reported associations between FPG and TG levels and postoperative complications in abdominal surgery. The pathological mechanism of glucotoxicity affecting the function of the circulatory system involves hyperglycemia and its secondary oxidative stress. Elevated FPG and TG levels result in increased glycosylated hemoglobin, which exerts pro-inflammatory and thrombotic effects, potentially leading to postoperative complications (Daryabor et al., 2020; Libby, 2007). For the pancreas, insulin resistance can lead to dysfunction of pancreatic islets β cells, resulting in persistent hyperglycemia (Chinese Elderly Type 2 Diabetes P et al., 2022). The TyG-BMI index, which is calculated using TG, FPG, and BMI has become an attractive option for predicting insulin resistance. Several studies have shown that the TyG-BMI index is associated with inflammation, glucolipid metabolism disorders, and microvascular complications (Yang et al., 2023). However, the correlation between the TyG-BMI index and complications after PD remains unclear. Thus, this study aimed to investigate whether the TyG-BMI index can predict major complications in patients undergoing PD.

SIRI is a novel systemic inflammatory biomarker based on the counts of neutrophils, monocytes, and lymphocytes in peripheral blood, which can reflect the immune and inflammatory status in patients. It was initially used to predict the survival of patients undergoing chemotherapy for pancreatic cancer (Qi et al., 2016). It can also be used to predict the risk of postoperative complications, assess postoperative recovery, and guide postoperative interventions. This study found that patient group with major complications had higher SIRI. The findings may be explained by the following mechanisms. Monocytes promote inflammatory response by releasing pro-inflammatory cytokines and interacting with other immune cells, thereby increasing the risk of postoperative complications.

PNI is an important indicator reflecting the nutritional status of patients and is of great significance for perioperative nutritional assessment. PNI, as a combined indicator of serum ALB and lymphocytes, effectively enhances the sensitivity of their predictive power. Its predictive effect on the prognosis of malignant tumors of the digestive tract, such as pancreatic cancer and gastric cancer, has been reported in many studies (Huang et al., 2023; Pan, Ma & Dai, 2023). Especially in pancreatic cancer patients, the preoperative nutritional status is crucial for postoperative complications and prognosis (Tumas et al., 2020). Due to factors such as tumor location and malignancy, most pancreatic cancer patients develop symptoms related to malnutrition, with unintentional weight loss being the primary manifestation. Although various nutritional assessment tools are currently available and widely used in hospital, such as the Nutrition Risk Screening 2002 (NRS 2002) and the Malnutrition Universal Screening Tool (MUST), their results are often influenced by subjective descriptions and subjective judgments from evaluators (Hersberger et al., 2020; Zhang et al., 2021). In contrast, PNI, as an objective calculation formula based on laboratory tests, shows significant potential in the preoperative assessment of patients with cancer. This study indicates that patients with a low preoperative PNI are more likely to occur major complications. Therefore, using PNI to assess preoperative nutritional status and the risk of major complications can be beneficial for the long-term outcomes of patients.

Another risk factor for major complications in our model is CA199 ≥ 37. Serum tumor markers are a commonly used diagnostic and therapeutic tool in clinical practice, known for their convenience, cost-effectiveness, and repeatability. CA199 is the most commonly used serum tumor marker in the diagnosis and treatment of pancreatic tumors and cholangiocarcinoma (Engle et al., 2019). In healthy individuals, serum CA199 levels are typically less than 37 U/ml. Studies have shown that patients with elevated preoperative CA199 levels are more prone to postoperative complications and that it is associated with poor prognosis in various tumors (Cai et al., 2023; Wang et al., 2023; Zhang et al., 2022a). In this study, CA199 was found to be an independent risk factor for major complications after PD. The possible explanation is that in the tumor microenvironment, hypoxia induces tumor cells to proliferate extensively and produce more CA199, which is released into the bloodstream after cell destruction. Additionally, tumor infiltration promotes angiogenesis, increasing vascular permeability, making it easier for CA199 to enter the bloodstream (Galli, Basso & Plebani, 2013).

Therefore, our study developed a nomogram focused on predicting postoperative major complications in patients undergoing PD. In addition, we compared our model with existing new methods for predicting postoperative complications after PD. The AUC of our model is 0.854, which is higher than the AUC1: 0.723 (Zhu et al., 2022), AUC2: 0.701 (Shen et al., 2021b), AUC3: 0.821 (Li et al., 2019), and AUC4: 0.776 (Li et al., 2021). The results showed that our model has good predictive capability for major complications after PD. Additionally, there are also some limitations in this study that need to be considered. Firstly, although this study included two centers, it was a retrospective study and the sample size was small, so inherent bias was inevitable. Larger, prospective, and multi-center studies are required to confirm the efficacy of this research. Secondly, the risk factors in our prediction model were derived from routine laboratory tests, potentially lacking some important parameters from specialist examinations. Thirdly, due to the limited duration of laparoscopic PD in our centers, the method of PD in this research is the open approach.

Conclusion

In conclusion, this two-center study identified the preoperative CA199 ≥ 37, SIRI, PNI, TyG-BMI, and VAV/SAV as significant risk factors in the nomogram for predicting major complications after PD. The nomogram exhibited good performance in predicting risk factors with high degrees of stability and accuracy, enabling clinicians to carry out necessary treatment strategies early to improve patient outcomes in the future.

Supplemental Information

Supplemental Information 1 The flowchart of this study.

Supplemental Information 2 Distribution of postoperative complications.

Supplemental Information 3 The incidence of postoperative complications of Clavien-Dindo.

Supplemental Information 4 LASSO variable screening process.

Supplemental Information 5 LASSO model coefficients.

Supplemental Information 6 ROC curves of the training database and test database.

Supplemental Information 7 The calibration curve of training database.

Supplemental Information 8 The calibration curve of test database.

Supplemental Information 9 The DCA curve of the training database.

Supplemental Information 10 The DCA curve of the test database.

Supplemental Information 11 Raw data.

The lasso and multivariate logistic regression analysis results.

Additional Information and Declarations

Competing Interests

Author Contributions

Human Ethics

Data Availability

The authors declare that they have no competing interests.

Jiaqi Wang conceived and designed the experiments, performed the experiments, prepared figures and/or tables, authored or reviewed drafts of the article, and approved the final draft.

Kangjing Xu conceived and designed the experiments, performed the experiments, prepared figures and/or tables, authored or reviewed drafts of the article, and approved the final draft.

Changsheng Zhou performed the experiments, prepared figures and/or tables, authored or reviewed drafts of the article, and approved the final draft.

Xinbo Wang performed the experiments, prepared figures and/or tables, and approved the final draft.

Junbo Zuo analyzed the data, prepared figures and/or tables, and approved the final draft.

Chenghao Zeng analyzed the data, prepared figures and/or tables, and approved the final draft.

Pinwen Zhou analyzed the data, prepared figures and/or tables, and approved the final draft.

Xuejin Gao analyzed the data, prepared figures and/or tables, and approved the final draft.

Li Zhang analyzed the data, prepared figures and/or tables, and approved the final draft.

Xinying Wang conceived and designed the experiments, prepared figures and/or tables, authored or reviewed drafts of the article, and approved the final draft.

The following information was supplied relating to ethical approvals (i.e., approving body and any reference numbers):

This study conformed to the ethical guidelines of the 1975 Declaration of Helsinki and was approved by the Ethics Committee of Jinling Hospital (2023DZKY-049-03) and registered at ClinicalTrials.gov (NCT06378853).

The following information was supplied regarding data availability:

The raw data is available in the Supplemental File.

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
