# Peer review of "A novel model based on clinical and computed tomography (CT) indices to predict the risk factors of postoperative major complications in patients undergoing pancreaticoduodenectomy"

_PeerJ, doi:10.7717/peerj.18753_

## Round 0.1 · original submission · Major Revisions

The authors are requested to carefully revise the manuscript and answer the questions raised by the reviewers.

·

Basic reporting

The manuscript is fairly well written with reasonably satisfactory English throughout. There are however some ambiguities in statements as indicated in General Comments. The introduction and background provided appropriately contextualised the project. The manuscript is generally structured in line with PeerJ standards. So are the raw data provided. Figures and Tables may benefit from more better numbering system as stand-alone components of the manuscript. Suggested considerations for the Tables and Figures are addressed under General/Additional Comments.

Experimental design

The design demonstrated the originality of the project. The study may well be a first of its kind type into use of 3D CT reconstruction in this field. The research question is well defined, clear, meaningful and relevant to the clinical problem being investigated. The relevance of the research in current clinical practice is well described and demonstrated throughout. Appropriate methodology and statistical interrogations of the study is adequate

Validity of the findings

All relevant underlying data as provided are robust, seems complete and statistically meaningful. The conclusions are clear and stated in refence to the main research question which was to demonstrate a novel model based on 3D CT reconstruction to predict post operative complications post PD for pancreatic ductal adenocarcinoma. Although this project demonstrated a compelling clinical relevance for the nomogram, the full utility and usefulness will require appropriate prospective, multicentre, international validation studies.

Additional comments

This study is a well-conceived and executed project with sound methodology, appropriate statistical interrogations as well as size able number of cases for an initial foray into developing a clinically useful nomogram model for predicting post PD complications. Though data of PDAC patients were used, the model is likely to hold true for all PDs conducted for other indications.
Of note however, the manuscript and its message may be enhanced with considerations of followings:
i. Depersonalization of the manuscripts. Terms as ‘we’, ‘our’ etc could be de-personalized for better import of the communication. Depersonalization also ensures the error of “being too close” to one’s data and as such failure to dispassionately and critically interrogate or evaluate things. Lines 87, 113, 226, 276, 278, 299, 301, 314, 319, etc should be considered for rewording through an appropriate writing expert
ii. Tables and Figures: Authors to consider changing current manner of depicting the Tables and Figures to a more international nomenclature of 1, 2, 3,… etc so that each figure and table stand alone. Where necessary, a, b, c, …etc may be used for sub-tables or sub-figures.
iii. Line 74 – 77: The meaning of the point here is lost in the convoluted manner of writing. Requires clarifications.
iv. Line 81: “research”: provision of references of studies will be helpful
v. Line 104: The first part of this line belongs to the results section and should be relocated accordingly.
vi. Line 105: The word “prospectively” here contradicts the “retrospective” term used in line 26 under the abstract. Authors to clarify and amend as necessary
vii. Lines 108 – 113: The stated criteria could be obviated by a statement such as “all patients that underwent upfront PD during the study period at the designated institutions by expert pancreatic surgeons”. At any rate, the exclusion criteria are merely opposite of the inclusion criteria and need not be listed again. Seems totological.
viii. Line 115: Consider replacing with “in terms of the Clavien-Dindo system, the cohort was divided into those with or those without complications”.
ix. Line 126: the experience of surgeons adds little to the quality of the study. Leaving this out may be prudent
x. Lines 125 – 139: Full description of the surgical technique is probably adds little to the study as the research is not about the details of surgical procedures. The authors should consider a shorter version with a focus on followings:
a. Was the surgery by Laparoscopy or open
b. Was the initial approach an “artery-first” approach or “Kausch-Whipple” description
c. Brief information and reference about the National Comprehensive Cancer Network guideline. This is presumably the guideline from China. So, a reference will assist international readers to follow well
d. A full description of the reconstruction as per lines 134 – 139 is probably unnecessary as it adds little to the manuscript quality. Keeping the reconstruction on the types and how the construction of the anastomoses are more than enough. By anastomotic constructions, I mean either hand-sewn or not and whether by interrupted or continuous suturing techniques.
e. Later parts of line 139 starting with “Finally”… till line 141 require re-writing. Current version is as a form of command or directives.
xi. Line 142: state the type of drug in terms of whether they are Proton pump inhibotors or other forms.
xii. Line 144: Is this the routine of the study sites, or just for this study? If routine protocol of the Units, providing the reason will clarify things.
xiii. Lines 163 – 166: consider re-writing as “the maximum tumor size, tumor differentiation, length of hospital stay, …” were analysed.
xiv. Line 169: is the “trained investigator” a radiologist or just a researcher? If a radiologist, stating this provides better clarity.
xv. Lines 177 – 179: Authors should provide brief details about BMI an CA199 cut offs will assist international readers to better understand the point herein.
xvi. Line 199: Consider writing out the LASSO in full again, to improve readability
xvii. Lines 203 – 204: Provide references for this uncommonly used statistical analyses
xviii. Line 212: Consider re writing as “Of 245 PDs performed at the participating institutions during the study period, 204 (% in relation to 245) met the criteria for analysis”.
xix. Line 214: Consider re writing as ‘there were 103 males (%) and 101 females (%)”
xx. Lines 225 - 229: Feature Selection: These are methodology information. Also re-writing the points being made will enhance the import and importance of the message. “Age, gender, NLR, NAR, … were evaluated by the LASSO regression to produce 6 significant parameters (NLR, SIRI, PNI, …) inputted into the final multivariate regression analysis.
xxi. Line 230: may read as “The multivariate regression analysis conducted as described demonstrated that ……”
xxii. Lines 234 – 237: Similarly, the information contained in these lines is better under methodology, whilst the C-index results be described in results section.
xxiii. Discussion:
xxiv. Line 253: “Until now”, this seems ambiguous as PD is being most effective therapy for resectable PDAC. So authors to consider expunging this.
xxv. Lines 257 – 259: Unclear what the authors mean by this somewhat ambiguous statement. Perhaps what is meant was “All cause-morbidity rates after PD are reported to vary from 38 – 77%, whilst procedure specific complications vary between 13 – 50%”. If so, then consider re-writing.
xxvi. Line 262: Authors to provide reference/s of the “several studies” been mentioned

·

Basic reporting

Manuscript ID Submission ID 101934v1
This paper is related to reviewing the manuscript titled " A novel model based on three-dimensional CT reconstruction to predict the risk factors of postoperative complications after pancreaticoduodenectomy in pancreatic ductal adenocarcinoma patients"
In this study, a retrospective analysis of 204 pancreatic ductal adenocarcinoma (PDAC) patients who underwent pancreaticoduodenectomy (PD) assessed the association between adipose tissue volumes, systemic inflammation, and postoperative complications. Three-dimensional computed tomography measured visceral and subcutaneous adipose volumes, while the Clavien-Dindo classification graded complications. Logistic regression analysis identified visceral adipose volume, systemic inflammation response index (SIRI), triglyceride glucose-body mass index (TyG-BMI), and prognostic nutritional index (PNI) as independent risk factors for complications. A nomogram incorporating these factors demonstrated excellent discrimination (C-index: 0.812, AUC: 0.836) and strong clinical utility.

Experimental design

Firstly, Although the proposed study is successful in terms of organization, presentation, content and results, major revision given in the following items need to be performed.
1) According to the authors, a predictive model was constructed based on the results of least absolute shrinkage and selection operator (LASSO) and multivariate logistic regression analysis. However, the estimation method is not explained in the methods section.
2) The authors suggested using an artificial intelligence-based deep learning model for predicting of the pancreatic adenocarcinoma disease, however, neither the mathematical nor algorithmic expressions of these prediction methods are given in the paper text. The authors urgently need to find a solution to this issue, and the artificial network models and mathematical equations of the methods and deep learning must be given in the paper.
3) What are the contributions of the authors in this study in terms of computer science and artificial intelligence? It is essential to clarify this issue.
4) Although the sensitivity in the ROC curve in Figure 3 is almost 100%, a lower score of 0.8xx AUC was obtained in the same result. This situation proves the inconsistency in the results of the study.
5) In addition, the proposed model should be compared with new methods, from the results except Figures 4 and 5.
6) Performance analyses and results are very few and insufficient. Increasing the results and including more detailed analyses in the article would increase the value and scope of this paper.

Validity of the findings

Firstly, Although the proposed study is successful in terms of organization, presentation, content and results, major revision given in the following items need to be performed.
1) According to the authors, a predictive model was constructed based on the results of least absolute shrinkage and selection operator (LASSO) and multivariate logistic regression analysis. However, the estimation method is not explained in the methods section.
2) The authors suggested using an artificial intelligence-based deep learning model for predicting of the pancreatic adenocarcinoma disease, however, neither the mathematical nor algorithmic expressions of these prediction methods are given in the paper text. The authors urgently need to find a solution to this issue, and the artificial network models and mathematical equations of the methods and deep learning must be given in the paper.
3) What are the contributions of the authors in this study in terms of computer science and artificial intelligence? It is essential to clarify this issue.
4) Although the sensitivity in the ROC curve in Figure 3 is almost 100%, a lower score of 0.8xx AUC was obtained in the same result. This situation proves the inconsistency in the results of the study.
5) In addition, the proposed model should be compared with new methods, from the results except Figures 4 and 5.
6) Performance analyses and results are very few and insufficient. Increasing the results and including more detailed analyses in the article would increase the value and scope of this paper.

Reviewer 3 ·

Basic reporting

This study presents an innovative approach by constructing and validating a model to predict postoperative complications for patients undergoing pancreaticoduodenectomy (PD). The integration of 3D CT reconstruction and the identification of novel risk factors make a valuable contribution to the existing literature, helping to better understand the occurrence of morbidities. However, several aspects need to be addressed to improve the manuscript.

1. The C-index of the model is reported as 0.812 (95% CI: 0.698-0.794). Typically, the C-index should not exceed the upper limit of its 95% confidence interval. Please clarify or correct this discrepancy.
2. PD is not only a curative treatment for pancreatic ductal adenocarcinoma (PDAC) but also for other neoplasms such as intraductal papillary mucinous neoplasms (IPMN). Since this study does not focus on survival outcomes, the sample size could be increased by including patients who underwent PD for other tumor types. Additionally, the introduction could be revised to reflect the broader application of PD rather than focusing solely on PDAC, which is not directly related to the primary topic of the study.
3. The study uses the Clavien-Dindo classification (CDC) to assess complications, yet patients are simply divided into those with or without complications. This approach seems to render the CDC classification underutilized. In similar research, it is common to stratify patients into groups based on CDC III-V versus CDC 0-II, as the latter typically does not significantly impact recovery. To enhance the representativeness of your conclusions, consider implementing this stratification.
4. There appears to be a typo on line 88: “PCAD” should be corrected.
5. While the introduction discusses TyG-BMI and VAV/SAV, there is no mention of prognostic nutritional index (PNI) and systemic inflammation response index (SIRI), both of which play a key role in your study. Please provide background information on these factors and discuss their relevance in previous literature.
6. The section describing the PD surgical procedure is overly detailed. Consider summarizing it to focus more on the key aspects relevant to your study.
7. Several important factors influencing postoperative complications are missing from the data collection, such as past history of diabetes, hypertension, ASA score, and jaundice. Additionally, factors like pancreatic texture, pancreatic diameter, blood loss, and operative time, which are critical for predicting complications like postoperative pancreatic fistula (POPF), should be included. Please incorporate these variables into your analysis.
8. The formulas used in data collection should be appropriately cited from previous literature. Additionally, definitions for each complication, such as those based on the International Study Group on Pancreatic Surgery (ISGPS) criteria, should be provided since these complications are central to your study.
9. In Table S1, please include the proportion of CDC stages for each complication to provide a more detailed understanding of their severity.
10. The criteria for selecting only 9 features for LASSO regression are unclear. Why were other potential factors excluded? Please clarify the selection process and provide results for the factors included in the logistic regression analysis.
11. Previous studies have shown that common BMI can predict morbidities. Your study did not include BMI in the logistic analysis. Please discuss the advantages of using TyG and VAV/SAV over common BMI for predicting morbidities.
12. The discussion section includes extensive commentary on the complexity of PD and its postoperative complications, which is already well-established. The first and second paragraphs of the discussion seem unnecessary and could be removed. Instead, focus on the potential mechanisms by which the novel factors you identified may influence morbidities. The current discussion of these mechanisms is insufficient and lacks depth. Expanding this section will make the content more impactful.
Considering the points above, this manuscript does not currently meet the standard for publication. I encourage the authors to revise the manuscript in line with these suggestions.

Experimental design

no comment

Validity of the findings

no comment

Additional comments

no comment

---

## Round 0.2 · Major Revisions

The authors are requested to carefully revise the manuscript and answer the questions raised by the reviewers.

·

Basic reporting

Since the authors did not fully implement the corrections and suggestions requested in the first round, I do not find it appropriate for this manuscript to be published as an article in this journal.

Justifications: They did not increase or improve the results as requested, they compared them with a different new algorithm requested in the first round. I believe that the results obtained in this study do not meet the standard quality of this journal.

Experimental design

None

Validity of the findings

None

Reviewer 3 ·

Basic reporting

Thank you for the opportunity to review the report again. The authors have addressed all my concerns。

Experimental design

/

Validity of the findings

/

Additional comments

/

Reviewer 4 ·

Basic reporting

The article primarily constructs and validates a predictive model for major postoperative complications in patients undergoing pancreaticoduodenectomy (PD). Over five years, the study collected data from 360 patients and used 3D CT reconstruction to measure visceral adipose volume (VAV) and subcutaneous adipose volume (SAV). Postoperative complications were classified based on the Clavien-Dindo system. The predictive model was developed using the results of least absolute shrinkage and selection operator (LASSO) multivariate logistic regression analysis and stepwise (stepAIC) selection. The authors identified that high VAV/SAV, high systemic inflammation response index (SIRI), high triglyceride glucose body mass index (TyG-BMI), low prognostic nutritional index (PNI), and CA199 ≥ 37 are independent risk factors for complications.
The overall structure of the article lacks clarity, and the main theme is not emphasized enough. The significance of these features in the risk model needs to be better explained, but the title does not reflect the main topic. It is suggested that the author revise the title to highlight the risk model assessment rather than focusing on 3D CT reconstruction, as this may mislead readers into thinking that the study is primarily about 3D CT reconstruction. Additionally, according to the PeerJ journal's guidelines, the innovative aspect of the study is not prominent. The study follows standard methods and uses the subcutaneous adipose tissue (SAT) area in the paralumbar section of the third lumbar spine (L3) as a key focus for analyzing complications. Overall, there are still many issues with the details of the paper, but with revisions, it has the potential to be a good article.

Experimental design

1.The title of the article needs to be revised. From the Lasso results, VAV/SAV is significant, and it is well differentiated from PNI, TyG-BMI, which are not directly related to adiposity, and the largest coefficient is SIRI. The article should focus on analyzing these contents, and the title should be as close as possible to this direction.
2. Nomogram does not list the c-index and other relevant parameters to evaluate the prediction effect, and there is no interpretation of the results of the prediction model, such as the lack of Fig5 to describe the effect of optimization of the nomogram in 3.4.
3. Lasso results refer to figure S3, the interval between the dotted lines is large on behalf of the coefficient of reduction of large changes in the model is not concise enough, there may be multiple covariance between the variables, may also be due to the introduction of fewer variables lead to, it is best to attach a description in the article.
4. The method description part of the article, the part about the data collection can be appropriately reduced, the article focuses on modeling can be more in the method part of the modeling-related content, such as how to optimize the model instead of a sentence over.

Validity of the findings

The article focuses on subcutaneous adipose area in the paralumbar section of the third lumbar spine as a potential mechanism that may affect morbidity, and this feature was extracted by 3D CT reconstruction. The article makes good use of the nomogram prediction model to predict complications and prognosis after PD, and the article has clinical applications.

Additional comments

Overall, this study has limited innovation and uses a simple predictive modeling structure, mainly incorporating more research on the effects of subcutaneous adipose areas on postoperative complications, and the discussion section of the article has an in-depth study on the effects of these features on complications, but the title and abstract are a bit different from the article's topic, which can easily cause other readers to misunderstand the content of the article, which would have been more perfect if the article were revised in accordance with the above mentioned conditions.

---

## Round 0.3 · Minor Revisions

The authors are requested to carefully revise the manuscript and answer the final questions raised by the reviewer.

Reviewer 4 ·

Basic reporting

no comment

Experimental design

no comment

Validity of the findings

no comment

Additional comments

Through CT indicators (Visceral adipose volume, VAV and subcutaneous adipose volume, SAV) into clinical prediction models to predict major complications in patients undergoing pancreaticoduodenectomy (PD) has good clinical significance, and the title of the article has been modified to fit the topic. The methodology section of the article has also been modified according to the suggestions given, but some details need to be optimized such as:Line217: Really use “prospectively”?

---

## Round 0.4 · accepted · Accept

After revisions, two reviewers agreed to publish the manuscript. I also reviewed the manuscript and found no obvious risks to publication. Therefore, I also approved the publication of this manuscript.

Reviewer 4 ·

Basic reporting

no comment

Experimental design

no comment

Validity of the findings

no comment

Additional comments

The authors were sincere in modifying the relevant comments made, and the article has some clinical research inspiration, so I recommend the article for publication in PeerJ.